# A Direct Comparison, and Prioritisation, of the Immunotherapeutic Targets Expressed by Adult and Paediatric Acute Myeloid Leukaemia Cells: A Systematic Review and Meta-Analysis

**DOI:** 10.3390/ijms24119667

**Published:** 2023-06-02

**Authors:** Vanessa S. Morris, Hanya Ghazi, Daniel M. Fletcher, Barbara-ann Guinn

**Affiliations:** 1Department of Chemistry and Biochemistry, University of Hull, Kingston upon Hull HU6 7RX, UK; v.s.morris@liverpool.ac.uk; 2Hull York Medical School, University of Hull, Kingston upon Hull HU6 7RX, UK; hyhg7@hyms.ac.uk; 3Centre for Biomedicine, Hull York Medical School, Kingston upon Hull HU6 7RX, UK; d.m.fletcher-2018@hull.ac.uk

**Keywords:** acute myeloid leukaemia, immunotherapy, paediatric, adult

## Abstract

Acute myeloid leukaemia (AML) is characterized by impaired myeloid differentiation resulting in an accumulation of immature blasts in the bone marrow and peripheral blood. Although AML can occur at any age, the incidence peaks at age 65. The pathobiology of AML also varies with age with associated differences in incidence, as well as the frequency of cytogenetic change and somatic mutations. In addition, 5-year survival rates in paediatrics are 60–75% but fall to 5–15% in older AML patients. This systematic review aimed to determine whether the altered genes in AML affect the same molecular pathways, indifferent of patient age, and, therefore, whether patients could benefit from the repurposing drugs or the use of the same immunotherapeutic strategies across age boundaries to prevent relapse. Using a PICO framework and PRISMA-P checklist, relevant publications were identified using five literature databases and assessed against an inclusion criteria, leaving 36 articles, and 71 targets for therapy, for further analysis. QUADAS-2 was used to determine the risk of bias and perform a quality control step. We then priority-ranked the list of cancer antigens based on predefined and pre-weighted objective criteria as part of an analytical hierarchy process used for dealing with complex decisions. This organized the antigens according to their potential to act as targets for the immunotherapy of AML, a treatment that offers an opportunity to remove residual leukaemia cells at first remission and improve survival rates. It was found that 80% of the top 20 antigens identified in paediatric AML were also within the 20 highest scoring immunotherapy targets in adult AML. To analyse the relationships between the targets and their link to different molecular pathways, PANTHER and STRING analyses were performed on the 20 highest scoring immunotherapy targets for both adult and paediatric AML. There were many similarities in the PANTHER and STRING results, including the most prominent pathways being angiogenesis and inflammation mediated by chemokine and cytokine signalling pathways. The coincidence of targets suggests that the repurposing of immunotherapy drugs across age boundaries could benefit AML patients, especially when used in combination with conventional therapies. However, due to cost implications, we would recommend that efforts are focused on ways to target the highest scoring antigens, such as WT1, NRAS, IDH1 and TP53, although in the future other candidates may prove successful.

## 1. Introduction

Acute myeloid leukaemia (AML) is an aggressive cancer that occurs due to the clonal expansion of immature white blood cells, known as blasts [1]. The accumulation of blasts occurs due to cytogenetic and molecular abnormalities that impair myeloid stem cell differentiation within the bone marrow [2]. Due to the impairment of normal haemopoiesis, the increased proliferation can cause blood-related problems, such as anaemia and haemorrhage [3].

The frequency, pathophysiology, treatment approaches, and survival rates vary between adult and paediatric AML, with age being a major factor in the frequency of AML diagnosis. In patients below 65 years of age, there are ~1.3 cases per 100,000 compared to ~12.2 cases per 100,000 in those above 65 years of age [2]. The increased AML rates in the older population are likely caused by the random accumulation of somatic mutations over time. Additionally, age seems to be a factor in the type of AML that patients develop. AML in the vast majority of paediatric patients develops de novo, whilst in adult patients AML more frequently develops from myeloproliferative neoplasms or myelodysplastic syndromes [1]. Along with age, there is often an increase in the activation of oncogenes, such as FLT3 [4], inactivation of tumour suppressor genes, such as WT1 [5], aberration of epigenetic modulators, such as TET2 [4], and decreased immune response [1]. Survival rates differ significantly between adult and paediatric patients, with one study showing a 5-year survival of ~17% in adult patients over 60 years of age [6], while survival rates in paediatric patients were around 60–75% [1]. The capacity of older and younger adults and children to tolerate the rigours of cancer treatment may explain why to date adult and paediatric patients have been treated in quite independent ways. Despite the differences in pathophysiology and initial survival rates, one thing AML has in common is that both adult and paediatric patients are at a high risk of death through a combination of relapse and treatment-related mortality [1]. Current treatment methods that include both induction and consolidation therapy often fail to eradicate AML cells that are protected by the bone marrow microenvironment and lead to relapse (reviewed in [7]). Thus, a new approach for AML treatment is essential to try to minimise the death toll caused by relapse.

Recently there has been a shift towards antigen-targeting immunotherapy in the treatment of cancers, including AML, to trigger anti-leukaemic responses [8]. In 2007, a National Cancer Institute Immunotherapy agent workshop [9] was held to rank immunotherapy targets based on the likelihood of cancer therapy efficacy. This well vetted list with broad and substantive input from academia, industry, and the government identified a number of immunotherapeutic agents that were effective cancer vaccine components in preclinical models. A subsequent National Cancer Institute pilot project sought to prioritise cancer antigens [10] based on their potential to act as translated targets for therapy. The methodology for prioritization was based on the analytical hierarchy process (AHP) [11], a mathematical model that helps teams deal with complex decisions that involve human perception and judgement. It has been used in a variety of settings, including government, industry, healthcare, and education, and helps by breaking down complex problems into pairwise comparable subproblems that are easier to quantify even when people have very different areas of specialisation. The model for prioritizing cancer antigens was developed through the combined efforts of experts in the field, and they assigned weighting to each of the criteria they ranked as being important for cancer vaccine development. Although with some limitations, this allowed for a uniform approach to the assessment of potential immunotherapeutic targets and a way to rank the most promising. There are multiple immunotherapeutic targets on the list of cancer antigens that are relevant to leukaemia and show promise in clinical trials. For example, both Phase I and II trials of a vaccine targeting PR1, a peptide derived from the leukaemia-associated antigen proteinase 3 [12], have been successful [13]. Recent vaccine developments have allowed for simultaneous targeting of multiple leukaemic antigens through the fusion of patient-derived AML cells to autologous dendritic cells, which induced an expansion of tumour-specific T cells, leading to extended remission times [14].

We wanted to determine whether the molecular pathways affected in AML were the same in adults and paediatrics, and whether the same immunotherapy targets could be used to treat AML across age groups. AML is diagnosed by molecular findings as described in the 2016 revision of the World Health Organisation (WHO) classification of myeloid neoplasms and acute leukaemia [15] which were published in the “blue book” monographs in 2001 and 2008, as third and fourth editions, respectively. However, we also wanted to include earlier articles that had used the French–American British (FAB) system of subtyping AML [16] and so we focused on studies of AML patients without cytogenetic abnormalities (such as t(8;21), inv(16) or t(15;17)), the FAB M4 subtype of AML known as acute myelomonocytic leukaemia, which has the highest prominence in the UK, and FAB M5, which includes acute monoblastic/monocytic leukaemia [17]. Our study examined all the available literature on antigen expression in paediatrics and adults with AML, since the inception of the literature databases until the current time. We have included a detailed meta-analysis that examines the commonality and differences in antigen expression and the pathways involved, and delivers a prioritisation of the top 20 antigens in each age-group as targets for immunotherapy clinical trials. By doing this we have established a large overlap in immunotherapeutic targets for adult and paediatric AML, illustrating the potential for a unified treatment approach.

## 2. Methods

### 2.1. Systematic Review

Standard systematic review methods were used and reported according to the PRISMA guidelines [18], including the development of a protocol (Appendix A). Google Scholar was used to ensure that this type of review of current research had not been undertaken recently, and the search strategy was developed based on index terms found in three to six sentinel articles that were identified in an initial screening of the literature using PubMed. The articles were identified using MeSH terms in articles, without limitation on when they were published. We wanted to show that the search terms identified relevant articles and that their own key words were encompassed by our search terms.

The study inclusion criteria were composed of both study and report characteristics (Table 1). The PICO system was used to design the research question, as follows: population—AML patients, intervention—expression of antigen targets in patient samples, comparison—expression in samples from healthy donors, outcome—potential as targets for therapy.

### 2.2. Search Strategy

Two independent reviewers (V.M. and H.G.) performed the search of the literature and screening process. Any disagreements were then settled by a third independent reviewer (B.G.) as and when required. MeSH terms and Boolean operators were used to search five databases (Scopus, the Cochrane library, CINAHL, MEDLINE, and PubMed) (Table 2) from inception until 4 March 2023. To ensure that all the relevant literature was used, alternative spellings of certain terms, such as leukaemia, were used. Studies using non-human models, animal models, comments, opinions, protocols, and non-English language publications were excluded.

### 2.3. Study Selection

The studies found within each database were exported into a single page of a Microsoft Excel Office 2019 sheet, combined, and duplicates were removed (Appendix A). The remaining articles were screened by title and then abstract, and articles were excluded if they did not meet the inclusion criteria. An in-depth screen of the articles was then performed. At the stage of abstract reading the search excluded books, systematic reviews, meta-analyses, and conference papers. The assessment of whether an article met the inclusion/exclusion criteria was performed by two independent researchers (V.M. and H.G.) and each reviewer were blinded to each other’s exact decisions until screening was complete. Any uncertainties in respect to the relevance of content were discussed with B.G.

Review articles were only removed once the cited articles in all selected manuscripts had been screened against the inclusion/exclusion criteria, as detailed above. This “backward snowballing” step helps ensure that relevant literature is successfully found as part of a systematic review [19].

### 2.4. Quality Assessment

A study specific version of QUADAS-2 [20] was applied to all articles at the final stage to factor in the risk of bias (Appendix A). Any articles that had a significant risk of bias were then excluded. Two independent reviewers (V.M. and H.G.) were used to minimise the risk of reviewer bias, and any disagreements were resolved through the involvement of a third reviewer (B.G.).

### 2.5. Data Extraction and Synthesis

From the final list of articles after quality assurance and risk of bias analysis, the potential targets of interest were separated into two data collection files, one for the antigens found in adult AML tissues and one for the antigens found in paediatric AML tissues (Appendix A). The NCBI number for each target was listed, as well as which articles it was listed within. Cancer antigens were then prioritized using the system devised by Cheever et al. [10], developed to rank vaccine targets based on predefined and pre-weighted objective criteria. The criteria in descending order of importance (followed by weighting in parenthesis) were: (a) therapeutic function (0.32); (b) immunogenicity (0.17); (c) role of the antigen in oncogenicity (0.15); (d) specificity (0.15); (e) expression level and percent of antigen-positive cells (0.07); (f) stem cell expression (0.05); (g) number of patients with antigen-positive cancers (0.04); (h) number of antigenic epitopes (0.04); (i) cellular location of antigen expression (0.02). None of the antigens were expected to have all the characteristics of an ideal antigen, but had one been perfect it would have scored 1.0.

### 2.6. Meta Analyses

PANTHER was used to illustrate which pathways are affected by events within the genome through predictions of gene interactions/functions. The 20 potential targets with the highest prioritization scores for both adult and paediatric AML underwent PANTHER analysis independently to produce a bar chart of the affected pathways.

STRING was also utilized to analyse the 20 highest scoring targets in both adult and paediatric AML separately. This created networks of known and predicted protein–protein interactions within *Homo sapiens* from a wide range of sources for both diseases. For both networks, the confidence levels of interactions were set to 0.7 to increase the accuracy of the network without excluding predicted connections. The database has in-built quality control and provides enrichment analysis to ensure the accuracy and relevance of the networks produced [21]. The identified inter-relationships can suggest which interactions define the behaviour of the biological system [22].

## 3. Results

### 3.1. Study Selection and Quality Assessment

The search strategy used identified a total number of 8577 articles (Appendix A). Of these, 5785 remained after the removal of duplicates. Articles were discarded in stages according to the exclusion criteria as shown in the Preferred Reporting Items for Systematic Reviews and Meta-Analyses (Prisma-P) flow diagram (Figure 1). A total of 36 articles remained at the end of the process, which were then subjected to QUADAS-2 assessment to check for risk of bias [20]. The QUADAS-2 scores (Appendix A) were then applied to each of the 71 potential targets (Table 3) which led to the elimination of eight targets from the final list of potentially viable targets for immunotherapy.

### 3.2. Data Extraction

A total of 90.7% of the genes identified as altered in paediatric patients were also identified in the literature describing altered gene expression in adults with AML (Figure 2; Appendix A). However, a further 21 genes were identified as having altered expression in adult AML tissues only.

### 3.3. Prioritisation of Tumour Antigens as Targets for Immunotherapy

We identified and assessed 63 targets and found that they ranged in score from 0.85 (WT1) down to the lowest scoring target at 0.17 (VEGFR-2) (Figure 3; Appendix A). A total of 20 antigens scored above 0.7, with all scoring highly for immunogenicity and most scoring low for expression levels and % positive cells. Overall, despite the highest value, 1, being assigned to the majority of targets for the number of epitopes identified due to each category being weighted differently, this category did not significantly contribute to the overall ratings.

### 3.4. Pathway and Protein–Protein Interaction Analysis

To determine which pathways were predominantly affected within adult and paediatric AML cells, the NCBI numbers of the top 20 targets, as identified by the modified Cheever model (in Section 3.3), were independently entered into Protein Analysis Through Evolutionary Relationships (PANTHER) [59]. PANTHER contains 176 different pathways within its database, of which 25 pathways were found to include the AML antigens identified in this study. There were notable similarities in the pathways involved as well as their relative involvement in AML between adult and paediatric patients (Figure 4). Only one pathway, the Alzheimer’s disease presenilin pathway, differed between the two age groups of AML patients, as it was only present within the top 20 for paediatric AML. The two pathways with the highest number of targets involved in both adult and paediatric AML age groups were angiogenesis and inflammation mediated by the chemokine and cytokine signalling pathway.

Inputting the top 20 targets into the Search Tool for the Review of Interacting Genes/Protein (STRING) for paediatric and adult AML patients showed their inter-relationships (Figure 5). STRING networks contain different coloured connections corresponding to the type of prediction; for example, experimentally determined interactions are connected in pink. This allowed for not only the number of different interactions to be determined but also for the strength of current evidence available to be indicated. The confidence levels of predicted interactions were set to 0.7 so that only highly likely interactions were identified. BIRC5 had the most relationships in paediatric AML, while PTPN11 had the most in adult AML.

## 4. Discussion

The databases selected meant that a large number of articles were systematically reviewed. This is because the Cochrane Library includes articles from the CINAHL database and MEDLINE^®^ includes results from PubMed. It is possible that relevant studies may have been missed if they were not written entirely in English or if the title or abstract did not make it clear that the outcome of the article was focussed on antigen expression in AML. However, the use of “reverse snowballing”, whereby citations in the selected articles were also subjected to the systematic review process, was used to mitigate this risk [19].

The databases used for the literature search have built in quality assessment. For example, Scopus openly advertises “a rigorous evaluation and selection process to ensure it meets the high-quality title selection criteria required for acceptance” and that journals “must demonstrate their ability to maintain quality status each year” with a “multi-step re-evaluation process” [60]. Additionally, the use of two independent reviewers combined with a third to settle any disputes allowed for an extra level of quality assurance. Before data extraction took place, the articles chosen from the initial searches underwent a quality assurance (QUADAS-2) step to ensure all included antigenic targets that achieved a sufficient quality of evidence to support their further analysis [20]. QUADAS-2 has two separate components to it: one focuses on the risk of bias while the other focuses on applicability to the study. Therefore, each of these components were taken into account when including or excluding studies. For example, if the patients studied had AML M6, they would have a low rating in the applicability section of the assessment. After assessing all selected papers using QUADAS, the total number of antigenic targets dropped from 72 to 63.

The criteria used excluded animal models. Traditionally animal models, predominantly rodent, have been utilized to investigate the underlying mechanisms of biological processes in a multiorgan setting, and can avoid some of the ethical and practical concerns that are associated with human studies. However, there are several drawbacks to the use of animal models within cancer research, which has called for a shift in how these models are used. The average success rate within clinical trials of the translation from animal models and patients is stated to be less than 8% [61]. This illustrates the severe limitations of animal models to mimic human carcinogenesis, pathophysiology, and disease progression. There are new methods being investigated to mitigate the differences seen between human and rodent models, including humanized mouse models and even humanized larger animal models of cancer [62,63]. Another potential limitation of animal models is their reduced life span that may fail to mimic human AML where age is a significant factor in both patient survival and treatment choice. Thus, it would be much harder to distinguish between adult and paediatric targets within animal studies, because of the very short time rodents remain paediatric.

Significantly fewer articles described the testing of paediatric patient samples, probably reflecting their relative scarcity compared with adult AML, and this may have contributed to the lower number of targets identified in younger AML patients. In addition, the reduced number of articles describing genetic change in paediatric AML may reflect historical concerns with the use of samples from paediatric patients in clinical trials due to the age of consent being 18 years old. However, it was recently stated that treatment intended for paediatrics should only be tested on that age group due to pathophysiological differences between adults and children [64]. In addition, survival rates for paediatric AML are much higher than for adult AML patients, which in addition to the elevated frequency of adult AML, makes adult AML more accessible to studies, while the need for new therapies, particularly for older adults, is a priority area for research. Some articles found in this study that had examined paediatric AML patients included patients in their late teens only (>15 years of age) [29,52]. This extremely narrow age-range could have skewed the results obtained as older paediatric patients have sometimes been observed to have an adult AML profile [65]. Most of the targets affected by this, such as WT1, also appeared in other articles from the literature search, mitigating this issue.

There were a vast number of similarities seen between adult and paediatric AML with 90.7% of antigens in paediatric samples also being described in adult AML. In fact, only four targets identified in paediatric AML were not shared with adult AML (CD44, CLEC12A, DARS-AS1, and TSC2). The large number of similarities observed between adult and paediatric AML supported the possibility of treatment repurposing, such as targeting BIRC5 (survivin) that had high expression levels in both adult and paediatric AML [66,67]. BIRC5 has been linked to multiple biological pathways including cell proliferation, angiogenesis, and metastasis [68], while above median levels of BIRC5 were associated with worse survival in adult AML patients with inv(16) [38]. FL118 targeting the BIRC5 promotor has been linked to transcriptional inhibition, and anti-tumour activity, in multiple cancer types, including cervical and pancreatic cancer and leukaemia [68,69,70].

The Initial extraction of data identified a large amount of targets; therefore, we chose the Cheever et al. [10] model of antigen prioritization to identify which antigens we would recommend for further studies. The Cheever model, however, does have limitations in that it has not been updated since its first publication in 2009. This means that there may be new important factors that the model did not take into consideration that may alter the relative weighting of each category. There is unfortunately no recent update on this or any other model that can give a predefined system for the ranking of potential targets. One example of a limitation that a more updated version may have mitigated was that the therapeutic function category focused on vaccine development while a wider range of immunotherapeutic approaches may be more appropriate. This led to some targets that have clear evidence of efficacy in other immunotherapy methods, such as combinational CAR T cells or tandem diabodies, (not vaccines per se) being scored lower despite having high therapeutic function. The biggest limitation of the scoring system was that there was no way to give novel antigenic targets the same weighting as those that had been previously investigated, thus, prioritising the more investigated rather than the more promising targets. Despite its limitations, the Cheever model of antigen prioritization offered a way to determine which antigens had the most promise as vaccine targets by considering rigorously determined criteria, and their relative weighting as set out by a panel of experts within the field. One study showed that the median cost of developing a single cancer drug was $648 million [71]; thus, it is clear that due to cost limitations only the targets with the most potential can be prioritized for development by funding agencies. Therefore, the method of scoring potential targets by Cheever et al. needs to be considered when deciding to move forwards with a target. However, to ensure the inclusion of the most promising targets for approaches other than just vaccines, the scores cannot be solely relied upon to make that decision. One example of this would be that CD123, which fell outside of the highest scoring targets after Cheever, has been used to target CLL in combination with CD3 via dual-affinity retargeting antibodies (DARTs) [72,73]. This suggests it has a higher immunotherapeutic potential than its original score for AML if the response initially seen in this CLL trial was to translate into AML patients.

Looking at both the amount of the literature that focussed on it and its top position after prioritization, WT1 was the most promising immunotherapeutic target for AML patients. This was the same outcome found by Cheever et al. [10] in a study that focussed more on data from patients with solid tumours by virtue of their predominance in the literature and in clinics. There has been previous success in preventing AML relapse through TCR targeting of WT1 with 100% relapse-free survival observed after 44 months [74]. However, this study only included adult patients and was implemented post-transplant. WT1 is an established target, but not novel. To identify a novel target, it may be best to rule out the therapeutic function part of the prioritization to instead focus on potential targets, not established ones.

CD13 could prove a viable target due to the high scores it achieved in key categories, such as immunogenicity [75] and the number of patients whose AML blasts express it [76]. One study investigated the influence of CD13 on the growth and survival of AML cells in vitro and found that CD13 could serve as a target for inducing caspase-dependent apoptosis within AML [77]. Despite being an in vitro study, these findings confirm that CD13 has the potential to be an effective target for a unified treatment approach. Additionally, Ubenimex is a CD13 inhibitor that was able to show promising effects from in vitro studies to clinical trials; in the Phase I clinical trials it was observed to prolong survival of AML patients but to also promote graft versus leukaemia effects in patients post-transplant [78]. Similar findings have been observed through targeting CD13 and TIM3 with bispecific and split CAR T cells to eradicate AML with reduced toxicity to the stem cells in preclinical models [75]. Therefore, despite CD13 not initially scoring as highly in the Cheever model as other potential targets, possibly due to the model’s focus on vaccine development, CD13 would be a very promising target for future therapeutics studies.

PANTHER was used to determine which pathways were affected by the 20 highest scoring targets in both adult and paediatric AML patients independently. In addition to paediatric AML sharing 96% of their immunotherapeutic targets with adults, both adult and paediatric AML have the most targets involved in angiogenesis and inflammation mediated by the cytokine and chemokine pathway(s). Angiogenesis is the generation of new blood vessels, which are key for cancer survival through the fresh supply of oxygen and nutrients to cancer cells [79]. Usually, angiogenesis decreases with age due to vascular ageing [80], so to have angiogenesis equally involved in both adult and paediatric AML was surprising. Two targets implicated in angiogenesis for both adult and paediatric AML were WT1, a key angiogenesis regulator [81], and BIRC5 [82]. Expression levels of BIRC5 varied depending on AML subtype but, within M4 and M5, the focus of this review, they were elevated. Even within M4/5 there was a potential issue with angiogenesis targeting other pathways, such as the platelet-derived growth factor (PDGF) signalling pathway, which can provide “escape” mechanisms, allowing cancer to resume growth [83]. Therefore, the most effective immunotherapy approach may be to simultaneously target multiple antigens and/or their pathways to limit the chance of relapse due to the escape of drug resistant clones [84].

Inflammation mediated by the cytokine and chemokine pathway was involved in both adult and paediatric AML development and had an equal number of targets involved in angiogenesis. Cytokines that are secreted within the bone marrow microenvironment are crucial in modulating cell survival, proliferation, differentiation, and the immune response [85]. The exact role of inflammatory cytokines in leukaemic cell expansion and the progression of AML are not yet fully understood; however, previous studies have indicated that pro-inflammatory cytokines contribute to an inflammatory microenvironment and have a growth-promoting effect on AML [86]. Both CXCR4 and JAK2 are linked specifically to inflammation via cytokines and chemokines, and are expressed by adult and paediatric AML cells, but JAK2 scored higher during prioritization. Chemokines’ primary function is the coordination and recruitment of immune cells and, therefore, they have a vital role in tumour development, aiding the formation of immunosuppressive and protective tumour microenvironments [87]. There has been previous interest in targeting cytokines and chemokines for immunotherapeutic treatment. Interferon-α was first approved for use on hairy cell leukaemia in 1986 and has since been used for the treatment of several haematological malignancies at high doses to exploit their pro-apoptotic activity on cancerous cells [88]. However, due to its relatively low efficacy and high cytotoxicity, its clinical use is now being reduced. Other cytokines and chemokines show promise, such as the anti-inflammatory mediators TGF-β and IL-10, which appear to impede AML progression [89,90].

The adult AML STRING network was created using the 20 highest scoring targets after Cheever prioritization to ensure the network only included the most relevant targets. TP53 was the target that displayed the most connections within the adult network. It has been predicted that ~10% of AML patients have mutations in TP53, which are associated with poor prognosis [91]. TP53 mutations are associated with older age in AML, affecting up to 25% of elderly individuals [92], and were within the top five ranked targets for both adult and paediatric AML. However, clinical trials targeting p53, directly or indirectly, remain a significant unmet need [93]. TP53 has been examined as a target for the treatment of AML using microRNA (miRNA) therapy, including miR-223, which is currently in preclinical trials. Despite there being little data on miRNA therapy in AML, the fact that miR-223 has been linked to all FAB subtypes does give promise that, with the correct delivery system, it could provide an effective targeted treatment for a wide range of patients [94]. Similarly to other targeting techniques, toxicity issues would need to be addressed, which is where both delivery system and potential combinational approaches would need careful consideration.

In paediatric patients, STRING indicated that the target with the most connections was PTPN11. PTPN11 also had a lot of evidence types within the STRING network which supports its potential success as a target despite not being frequently represented within this literature review. PTPN11 was found to be mutated in both adult and paediatric patients with AML and, therefore, could provide a unified approach. Mutations within PTPN11 were found in ~12% of adult AML patients and only 4% of paediatric AML patients [95,96]. Recent clinical trials illustrated that, despite the relatively low number of patients with mutated PTPN11, carrying the mutation was strongly associated with adverse clinical outcomes and understanding the mechanisms that caused this affect may facilitate the development of treatments [97]. Due to PTPN11′s low mutation rates, especially in paediatric patients, it may be beneficial to consider a combinational approach if it were to be selected for development. This would entail pairing it up with a more common target to try and increase the treatment’s overall efficacy so that it does not have to be ruled out entirely based on its infrequent subversion in AML alone.

In concurrence with PANTHER, STRING networks showed vast similarities between adult and paediatric patients when considering the top 20 antigenic targets in each age group. In adults, BMI1, CA9, PRAME, and TERT only featured in adults with AML, while CD44, CD244, CLEC12A, and IL-3RA only featured in paediatrics. One of the most connected genes within the top 20 for paediatric AML was JAK2, which, as previously mentioned, is part of the cytokine-mediated inflammatory pathway. JAK2 has been previously utilized as an immunotherapeutic target for breast cancer; however, there is little evidence of its successful targeting in AML so far, although it could provide a novel therapeutic approach [98,99]. This is possibly because JAK2 mutations are seemingly rare in AML, with 11 out of 339 AML patients having JAK2 mutations [100]. Thus, despite JAK2 mutations being associated with de novo AML [101], they may not prove a viable target in terms of broad applicability. Another promising target within the top 20 in both adult and paediatric AML was NPM1. This target not only scored highly within the Cheever prioritization model but also had a high number of connections within the STRING network. NPM1 was also evidenced as being crucial to AML development by a team who used CRISPR dropout screening to identify genetic vulnerability within five different cell lines [102]. This study illustrated that NPM1 was depleted in four of the five AML cell lines, which identified it as an AML-specific vulnerability [102]. One potential target that lies outside of the top 20 highest scoring targets was DNMT3A, which was found in a larger proportion of AML patients than JAK2, with 22.1% (62/281) of patients having alterations in DNMT3A [103]. The current literature also links DNMT3A and isocitrate dehydrogenase 1 enzyme (IDH1) to both adult and paediatric AML, which does suggest that their targeting could prove an effective approach across a broad range of age groups. Already, hypomethylating agents, such as azacytidine and decitabine, and IDH inhibitors, are the main therapeutic treatments for older AML patients who are unable to tolerate chemotherapy and relapsed/refractory AML patients harbouring an IDH mutation, respectively (recently reviewed in [104]).

From the information presented thus far, there are a number of potential targets identified for the therapy of AML; however, due to the high costs of developing treatments and the length of time that the approval of a treatment takes, it may be beneficial to focus on a few prioritised targets for treatment development over the next few years. In order to decide which targets should be selected, there are additional factors to consider including broad applicability across subtypes and age-boundaries. FLT3 mutations have been found in ~25% of paediatric patients and ~30% of adult patients [105], with some studies suggesting that successful targeting of unmutated FLT3 could lead to ~80% of AML patients being treated with one approach [106,107]. Previous targeting of FLT3 initially showed high promise as an effective treatment, but resistance to FLT3 inhibitors is already becoming a pressing issue that would need to be addressed for this to be effective long-term. Current ideas suggest that there are two forms of resistance (primary and secondary) and that these are caused by different underlying mechanisms [108]. One example of a primary resistance causing mechanism is that the upregulation of CXCL12 and FGF2 within the bone marrow microenvironment can shield AML blasts from FLT3 inhibitors [109,110]. Meanwhile secondary resistance has numerous causative mechanisms, but one of the major ones is epigenetic dysregulation caused by mutations in epigenetic modification genes including DNM3TA, TET2, and IDH1/2 [111]. A reduction in resistance to FLT3 inhibitors could be achieved through either the development of novel inhibitors or by considering a multitarget approach. It is also worth noting that recent publications demonstrated methods of expression amplification in FLT3. This has not yet been applied to AML, but, in breast cancer, hormone receptor antagonism was able to enhance the efficacy of targeted immunotherapy by amplifying target expression [112]. If this could be applied to AML, then it would allow for targets that may have lower expression to be utilized rather than focusing future work solely on targets with elevated expression levels.

Another key factor to consider when selecting which of the targets should be pursued are the toxicity issues caused by a lack of specificity. There are ways to mitigate this type of toxicity either through a combinational CAR T cell approach [113] or the development of built-in “safety switches” that would eliminate the CAR T cells after use [114]. With the advent of next generation immunotherapeutic strategies including Boolean gating, reversing T cell exhaustion, and split/SUPRA CARs (recently reviewed in [115]), we see an opportunity to address the current challenges limiting the safety and efficacy of CAR-T cells for cancer treatment. It is likely that a combinational approach can provide a safe delivery method for immunotherapeutic targeting of AML, and, indeed, recent studies suggest the combination of an immune checkpoint inhibitor and antigen targeting, as well as immune modulation, could further enhance anti-leukaemic responses [116,117].

## 5. Conclusions

AML is a rare but progressive form of blood cancer, which continues to have poor prognosis in both adult and paediatric patients. Low survival rates indicate a clear need for the development of new treatment approaches that will universally improve prognosis. The most promising approach are immunotherapeutic methods that target specific antigens that have a key role in the development and survival of AML cells. This could create an approach with higher efficacy and fewer side effects than high-dose chemotherapy alone. Adult and paediatric AML are currently treated as different conditions due to differences in underlying pathophysiology; however, immunotherapy offers one way to cross age boundaries by targeting common antigens. This review has identified antigens that could or do offer an opportunity to develop a unified treatment approach. Throughout all analyses in this review, there were significant overlaps between the antigens expressed in adult and paediatric AML, including within the most promising targets. A few examples of the overlapping targets that stood out at multiple stages were WT1, TP53, BIRC5, and JAK2, with some of these already being effectively targeted within clinical trials using a variety of delivery systems. However, there are a lot more, such as CD13 and PTPN11, that have been identified as potentially viable novel targets for a future unified approach. With the correct target selection and delivery method it may be possible to mitigate a lot of the side effects associated with the immunotherapeutic treatment of AML, such as toxicity caused by the lack of specificity. It is likely that there are ways to develop a unified approach of immunotherapy that could treat both adult and paediatric patients simultaneously and benefit from the cost savings that drug repurposing offers.

## Figures and Tables

**Figure 1 ijms-24-09667-f001:**
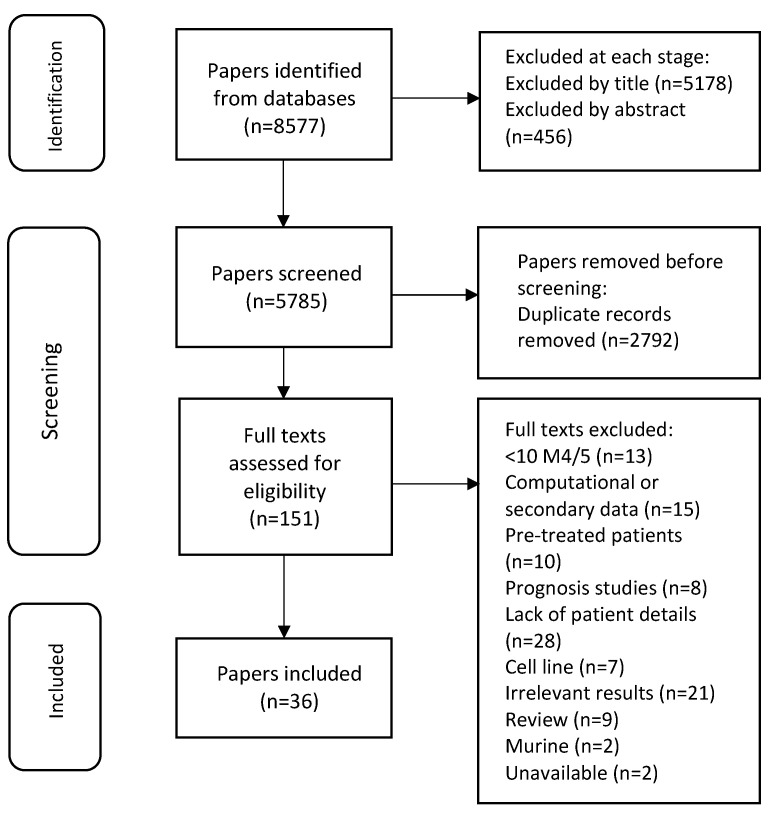
Prisma-P flow diagram to show the screening process that determined whether articles met the inclusion criteria at each stage; if not, they were removed. In total, 36 articles, describing 71 targets, were identified for the quality assurance step.

**Figure 2 ijms-24-09667-f002:**
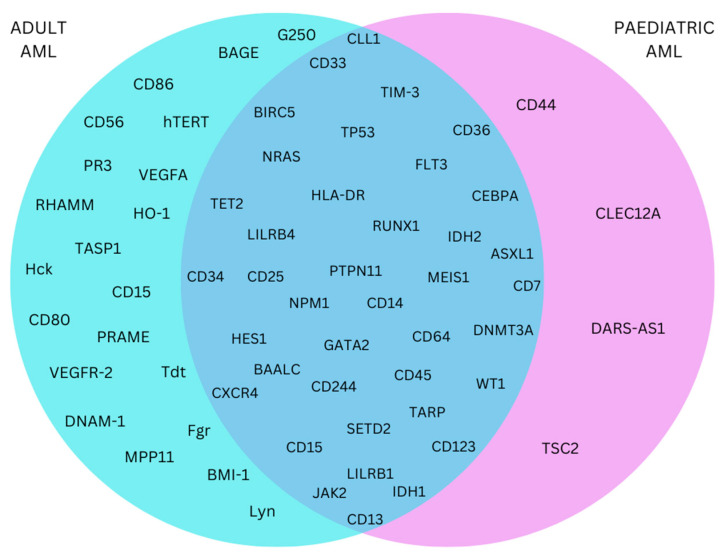
Venn diagram to illustrate which of the affected genes within adult and paediatric AML overlap with one another. Those that lie within the overlapping intersection should be assessed as potential targets for unified immunotherapy treatment approaches.

**Figure 3 ijms-24-09667-f003:**
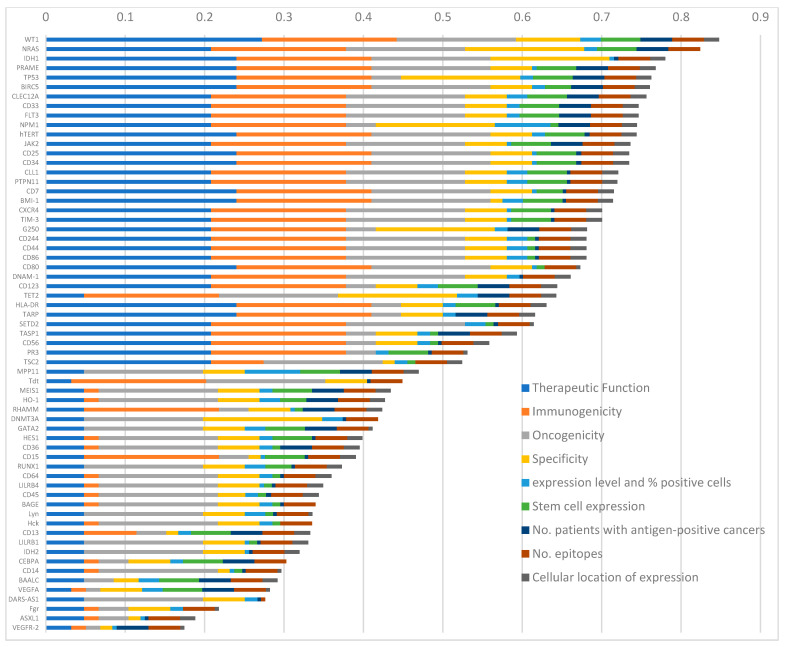
Prioritisation of the 63 potential treatment targets identified in AML by systematic review and found in articles that remained after quality control. Prioritisation was performed using the National Cancer Institute Pilot Project as a model [10], based on nine predefined, pre-weighted criteria. The antigens were scored to determine their suitability for further investigation as therapeutically effective cancer vaccines.

**Figure 4 ijms-24-09667-f004:**
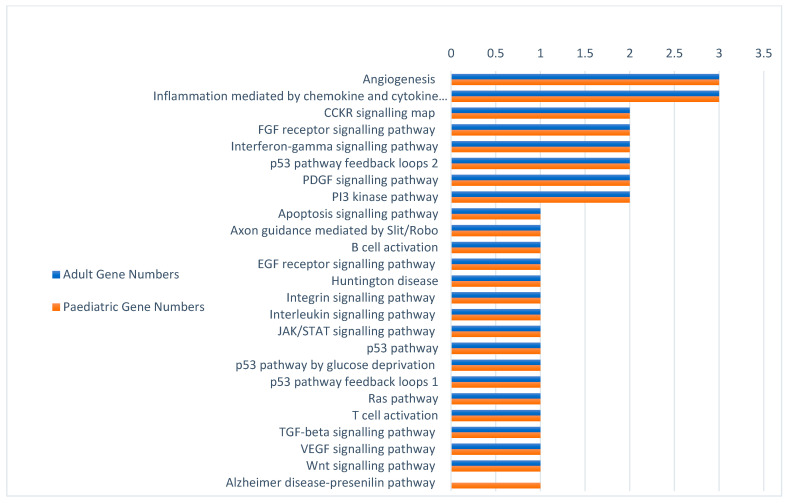
PANTHER was used to analyse the pathways that are linked to the 20 highest scoring targets after prioritization in both adult and paediatric AML. Pathways with the highest number of genes linked to them should be investigated further as key pathways for AML development. From those tested, only one pathway differs between paediatric and adult AML, which suggests strong similarities between the two and increases the possibility of an overlapping treatment.

**Figure 5 ijms-24-09667-f005:**
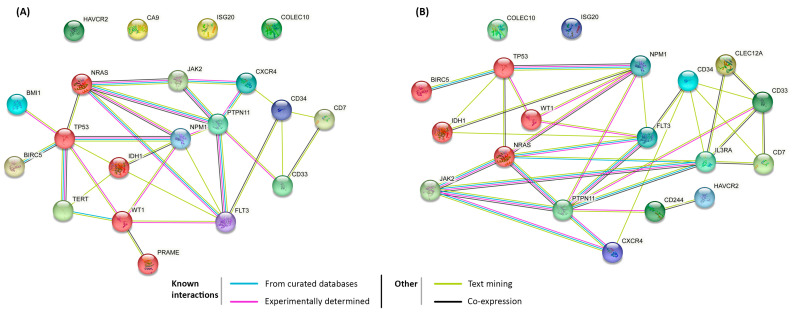
Protein–protein interaction analysis was performed via STRING with confidence levels set to 0.7 and the species set to *Homo sapiens*. In both figures, the top five targets (red nodes) are shown after prioritization for adult and paediatric AML. (**A**) STRING network of the 20 highest scoring potential therapeutic targets for adult AML identified after prioritization. TP53 had the most connections for the adult network. (**B**) STRING network of the 20 highest scoring potential therapeutic targets for paediatric AML identified after prioritization. PTPN11 had the most connections for the paediatric network.

**Table 1 ijms-24-09667-t001:** Inclusion criteria.

Category	Included
Publication date	All
Subgroup of interest	M4/M5 Acute Myeloid Leukaemia (FAB or WHO diagnosis)
Geographic location of publication	All
Language	English
Participants	All age groups, recently diagnosed
Reported outcomes	Antigen expression %
Study design	Over 10 participants
Publication type	Original research

**Table 2 ijms-24-09667-t002:** Publication database search strategy (Scopus example).

MeSH Terms and Boolean Operators Used to Search Terms	Filters
“acute myeloid leukaemia” OR “acute myeloid leukemia” OR “acute myelogenous leukaemia” OR “acute myelogenous leukemia” AND antigen OR “antigen therapy” AND human	Species: human
	Source type: journal
	Language: English
	Document type: article, review
	Age: adult and paediatric

**Table 3 ijms-24-09667-t003:** Risk of bias and quality assurance using the QUADAS-2 system of article assessment.

Study	Risk of Bias	Applicability Concerns
	PatientSelection	Intervention	Reference Standard	Outcome	Patient Selection	Intervention	Reference Standard
Dos Santos et al. [23]	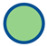	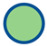	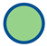	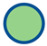	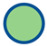	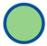	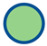
Xu et al. [24]	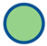	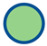	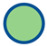	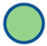	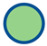	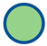	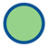
Darwish et al. [25]	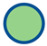	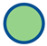	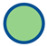	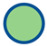	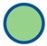	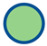	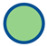
Xiao et al. [26]	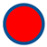	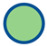	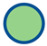	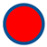	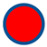	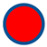	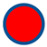
Wang et al. [27]	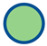	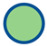	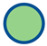	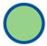	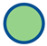	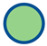	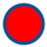
Xu et al. [28]	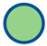	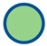	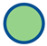	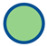	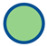	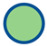	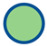
Haubner et al. [29]	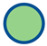	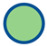	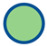	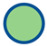	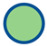	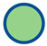	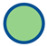
Abdellateif et al. [30]	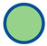	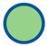	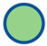	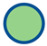	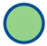	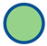	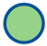
Vollmer et al. [31]	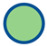	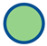	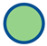	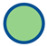	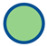	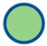	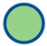
Zhu et al. [32]	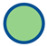	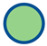	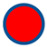	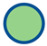	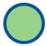	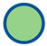	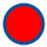
Lin et al. [33]	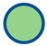	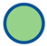	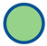	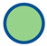	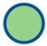	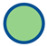	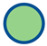
Tian et al. [34]	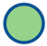	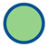	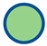	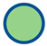	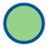	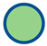	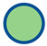
Xu et al. [35]	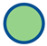	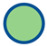	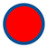	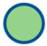	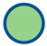	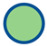	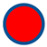
Basharat et al. [36]	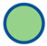	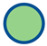	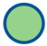	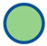	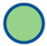	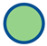	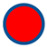
Alanazi et al. [37]	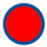	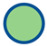	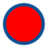	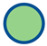	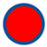	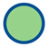	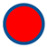
Greiner et al. [38]	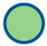	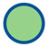	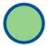	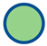	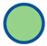	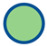	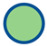
Shin et al. [39]	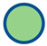	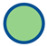	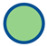	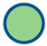	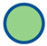	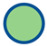	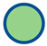
Dou et al. [40]	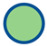	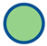	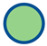	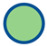	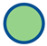	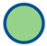	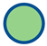
Padró et al. [41]	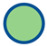	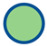	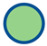	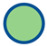	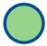	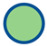	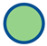
Assi et al. [42]	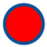	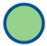	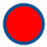	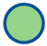	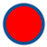	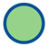	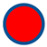
Venditti et al. [43]	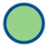	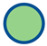	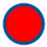	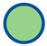	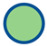	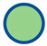	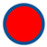
Djunic et al. [44]	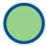	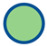	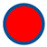	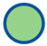	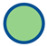	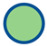	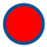
Du et al. [45]	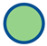	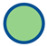	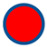	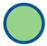	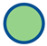	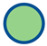	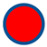
Aref et al. [46]	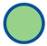	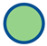	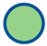	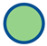	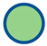	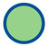	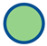
Plesa et al. [47]	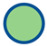	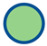	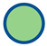	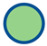	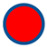	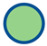	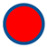
Cheng et al. [48]	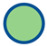	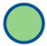	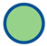	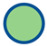	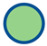	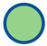	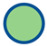
Lin et al. [49]	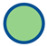	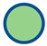	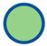	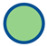	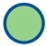	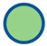	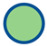
Zhang et al. [50]	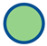	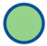	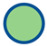	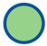	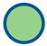	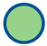	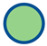
Depreter et al. [51]	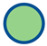	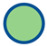	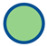	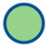	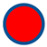	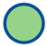	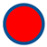
Rostami et al. [52]	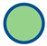	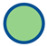	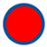	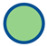	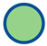	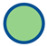	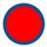
Steger et al. [53]	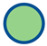	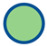	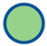	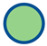	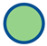	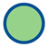	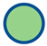
Willier et al. [54]	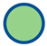	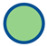	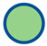	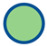	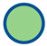	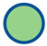	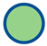
Chashchina et al. [55]	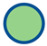	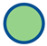	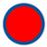	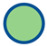	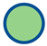	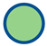	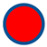
Churchill et al. [56]	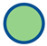	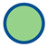	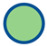	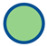	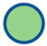	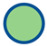	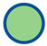
Wang et al. [57]	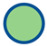	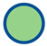	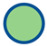	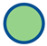	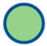	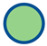	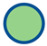
Sun et al. [58]	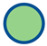	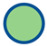	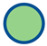	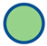	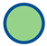	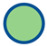	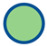

Red circles indicate a risk of bias; Green circles indicate no risk of bias, was identified.

## Data Availability

All data generated or analysed during this study are included in this published article and its Appendix A.

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
