# Peer review of "A Direct Comparison, and Prioritisation, of the Immunotherapeutic Targets Expressed by Adult and Paediatric Acute Myeloid Leukaemia Cells: A Systematic Review and Meta-Analysis"

_ijms, 2023, doi:10.3390/ijms24119667_

Round 1

Reviewer 1 Report (New Reviewer)

In this paper, the authors aim to present a systematic review on the most promising antigens for potential immunotherapeutic approaches for adult and pediatric AML. The aim is very ambitious since both pediatric and adult AML are considered. The general description of the work is well organized and presented.

In the abstract, specifically in the specific aim (lane 14-16) which refers to the ranking of antigens for their potential therapeutic significance, there is no reference to the fact that these antigens were ranked in the context of immunotherapeutical approaches. Also the title does not refer to the fact that the antigens were prioritized based on this treatment category.

In general, it is not clear why the authors decided to rank the antigens specifically for immunotherapeutical approaches rather than, or in addition to, other therapeutic approaches, could the authors please specify why this specific approach was taken into consideration for this specific study? Besides, immunotherapeutic approaches includes many different subcategories which substantially differs from each other, and for which antigens prioritization would probably need to be focused on specific target features, did the authors adjusted the ranking process for this aspect?

Did the authors took some important studies into considerations, such as the genome wide crispr screens conducted by Tselepis (Tzelepis et al. “A CRISPR Dropout Screen Identifies Genetic Vulnerabilities and Therapeutic Targets in Acute Myeloid Leukemia.” Cell reports vol. 17,4 (2016): 1193-1205. doi:10.1016/j.celrep.2016.09.079) and Wang (Wang et al. “Gene Essentiality Profiling Reveals Gene Networks and Synthetic Lethal Interactions with Oncogenic Ras.” Cell vol. 168,5 (2017): 890-903.e15. doi:10.1016/j.cell.2017.01.013), were many AML cell lines were screened, and from which many targets were listed and ranked as essential for AML? Did the authors considered the tractability of the targets?

Some more references to other previous meta-analysis considering AML antigens in the context of immunotherapeutic approaches may be added

It is not entirely clear why studies including work in animal models were excluded. This could lead to missing information regarding the antigens ranking

As the authors state in lane 298, the method to extract data is dated (2009): how did the authors compensated for this aspect? It seems that the data extracted does not show much novelty compared to what is already known about potential therapeutic targets to be used for immunotherapeutic approaches in AML

There are only few mistakes which need to be corrected

Author Response

Reviewer 1

In this paper, the authors aim to present a systematic review on the most promising antigens for potential immunotherapeutic approaches for adult and pediatric AML. The aim is very ambitious since both pediatric and adult AML are considered. The general description of the work is well organized and presented.

Thank you

In the abstract, specifically in the specific aim (lane 14-16) which refers to the ranking of antigens for their potential therapeutic significance, there is no reference to the fact that these antigens were ranked in the context of immunotherapeutical approaches. Also the title does not refer to the fact that the antigens were prioritized based on this treatment category.

This has now been specified on line 16 but is also mentioned on line 23 and the title changed to more accurately reflect the ranking of antigens on the basis of their ability to act as targets for cancer vaccines.

In general, it is not clear why the authors decided to rank the antigens specifically for immunotherapeutical approaches rather than, or in addition to, other therapeutic approaches, could the authors please specify why this specific approach was taken into consideration for this specific study?

This has been explained in line 261-264. The use of the Cheever score allowed us to determine the potential of an antigen to act as a cancer vaccine target. The Cheever scoring system was the first of its kind and has yet to be superseded.

Besides, immunotherapeutic approaches includes many different subcategories which substantially differs from each other, and for which antigens prioritization would probably need to be focused on specific target features, did the authors adjusted the ranking process for this aspect?

We focussed on antigens as targets for immunotherapeutic strategies rather than allowing the treatment strategy to limit the identification of antigen(s). Thus the chosen antigen(s) would inform the treatment strategy rather than vice versa.

Did the authors took some important studies into considerations, such as the genome wide crispr screens conducted by Tselepis (Tzelepis et al. “A CRISPR Dropout Screen Identifies Genetic Vulnerabilities and Therapeutic Targets in Acute Myeloid Leukemia.” Cell reports vol. 17,4 (2016): 1193-1205. doi:10.1016/j.celrep.2016.09.079) and Wang (Wang et al. “Gene Essentiality Profiling Reveals Gene Networks and Synthetic Lethal Interactions with Oncogenic Ras.” Cell vol. 168,5 (2017): 890-903.e15. doi:10.1016/j.cell.2017.01.013), were many AML cell lines were screened, and from which many targets were listed and ranked as essential for AML? Did the authors considered the tractability of the targets?

The Tzelepis et al study has now been cited and described to lines 378-381. However we excluded studies of cell lines and focussed on studies of primary patient samples because of the risks that cell lines often represent late stage disease and may have deviated (genetically) from the primary cell(s) of origin.

Some more references to other previous meta-analysis considering AML antigens in the context of immunotherapeutic approaches may be added

It is not entirely clear why studies including work in animal models were excluded. This could lead to missing information regarding the antigens ranking

We wanted to focus our analysis on human studies where antigens relevant to AML were identified, and their frequency of expression in the patient population determined. A paragraph has now been added to try and explain this in lines 205-218.

As the authors state in lane 298, the method to extract data is dated (2009): how did the authors compensated for this aspect? It seems that the data extracted does not show much novelty compared to what is already known about potential therapeutic targets to be used for immunotherapeutic approaches in AML

We did not compensate for the limitations of the Cheever et al, 2009 system of prioritising antigens, not least because the system was devised by a panel of experts, including a Nobel prize winner, and we wanted to compare our results prioritising AML antigens with those prioritised by Cheever whose study examined antigens from all cancers. To address the limitations we acknowledged them in lines 261 – 264 and 266-74.

Reviewer 2 Report (New Reviewer)

Acute Myeloid Leukemia (AML) is highly heterogeneous blood cancer, although the chemotherapeutic treatment can bring complete remission for majority patients, but most of them will develop relapse in next several years. In the current manuscript, the authors tried to determine whether the molecular pathways affected in AML were the same in adults and pediatrics and whether the same immunotherapy targets could be used to treat pediatric and adult AML. The authors identified common targets and specific targets for pediatric and adult AML. Overall, the authors provide some new thoughts and the conclusions may benefit our readership. the Reviewer only have one suggestion, since the pediatric and adult AML patients have huge differences, please discuss more.

Author Response

Acute Myeloid Leukemia (AML) is highly heterogeneous blood cancer, although the chemotherapeutic treatment can bring complete remission for majority patients, but most of them will develop relapse in next several years. In the current manuscript, the authors tried to determine whether the molecular pathways affected in AML were the same in adults and pediatrics and whether the same immunotherapy targets could be used to treat pediatric and adult AML. The authors identified common targets and specific targets for pediatric and adult AML. Overall, the authors provide some new thoughts and the conclusions may benefit our readership. the Reviewer only have one suggestion, since the pediatric and adult AML patients have huge differences, please discuss more.

Despite the huge clinical differences between adult and paediatric AML, the literature identified and antigens therein suggest that the pathways affected are often the same. Our study suggests that at least for some targets/pathways the same treatments could be used to target diseased cells in patients in either age group and that the age boundaries may, in fact, not be as distinct as previously thought. A few additional details on the differences seen in lines 50-53 and 60-66.

This manuscript is a resubmission of an earlier submission. The following is a list of the peer review reports and author responses from that submission.

Round 1

Reviewer 1 Report

I am sorry to report that this paper is frequently misleading and, in general, lacks crucial clinical knowledge of the AML field.

Line 38: infection is another major cause of death

45: each additional mutation in cancer-driving or tumor-suppressor

53: AML classification is nowadays based upon molecular findings. 

54: So far, in clinical trials, antibodies against AML cells have not been very successful, expecially when compared to B lymphoid malignancies where they are in backbone of almost every therapy.

89-97: Again, modern AML classification is molecular rather than the morphology-based FAB. I am not sure such an approach is nowadays acceptable 

182: What does "gene subverted" means? Mutated? Overexpressed? Silenced? Looking at Fig. 2, it shows a melting pot of mutated (eg IDH1-2) and overexpressed (eg VEGF) genes, some in AML cells and others likely in the microenvironment.

193: WT1, for instance, can be mutated (rarely) or overexpressed and used a biomarker. 

Fig. 3. I believe the legend generates confusion bteween druggable antigens and/or molecular targets. IDH1, for instance, is reported as the second most prioritized gene/antigen (confusion again..) but it is an iteresting target only in IDH1-mutated AML which are less than 10% of all AML.

Fig. 4: same story: albeit ranked here as #1 in prioritization, the angiogenesis pathway is unlikely to be an interesting clinical target, as witnessed by many clinical trials.

400: This sunitinib story is misleading: it has been developed as a PDGFR, and  VEGFR1, VEGFR2, inhibitor , with minor activity against KIT, and FLT.  The present generation of FLT3 inhibitors used in the clinic have a totally different specificity, efficacy and toxicity.

405: "mitigate toxicity" wirh CAR-T? These are very toxic in the clinic.

Round 2

Reviewer 1 Report

As reported by the Authors, "... our study does not determine clinical utility". In fact, it predicts utility for therapies that have not shown clinical benefit (eg anti-angiogenic therapies). I still believe this approach is misleading.

Reviewer 2 Report

I am grateful to you for the insightful clarification. The modifications that were made to the manuscript led to an increase in comprehension.